# A Derating-Sensitive Tantalum Polymer Capacitor’s Failure Rate within a DC-DC eGaN-FET-Based PoL Converter Workbench Study

**DOI:** 10.3390/mi14010221

**Published:** 2023-01-15

**Authors:** Dan Butnicu

**Affiliations:** Basics of Electronics, Telecommunications and Information Technology Faculty, Technical University “Ghe.Asachi” of Iasi, 700050 Iasi, Romania; dbutnicu@etti.tuiasi.ro

**Keywords:** derating, DC–DC converter, eGaN, PoL, conductive polymer tantalum chip capacitor, reliability, MTBF

## Abstract

Many recent studies have revealed that PoL (Point of Load) converters’ output capacitors are a paramount component from a reliability point of view. To receive the maximum degree of reliability in many applications, designers are often advised to derate this capacitor—as such, a careful comprehending of it is required to determine the converter’s overall parameters. PoL converters are commonly found in many electronic systems. Their most important requirements are a stable output voltage with load current variation, good temperature stability, low output ripple voltage, and high efficiency and reliability. If the electronic system in question must be portable, a small footprint and volume are also important considerations—both of which have recently been well accomplished in eGaN transistor technologies. This paper provides details on how derating an output capacitor—specifically, a conductive tantalum polymer surface-mount chip, as this type of capacitor represented a step forward in miniaturization and reliability over previously existing wet electrolytic capacitors—used within a discrete eGaN-FET-based PoL buck converter determines the best performance and the highest MTBF. A setup based on an EPC eGaN FET transistor enclosed in a 9059/30 V evaluation board with a 12 V input voltage/1.2 V output voltage was tested in order to achieve the study’s main scope. Typical electrical performance and reliability data are often provided for customers by manufacturers through technical papers; this kind of public data is often selected to show the capacitors in a favorable light—still, they provide much useful information. In this paper, the capacitor derating process was presented to give a basic overview of the reliability performance characteristics of tantalum polymer capacitor when used within a DC–DC buck converter’s output filter. Performing calculations of the capacitor’s failure rate based on taking a thermal scan of the capacitor’s capsule surface temperature, the behavior of the PoL converter was evaluated.

## 1. Introduction

Power electronics designers know that, when it comes to reliability improvement, voltage is one of the strongest accelerators for the occurrence of failure mechanisms. As such, its lowering may significantly improve the MTBF component of DC–DC converters’ output filters. The failure rate can be altered by the following parameters [1]:Voltage deratingTemperatureESR (equivalent series resistance)

In this practical study, it is shown that a conductive tantalum polymer chip capacitor’s MTBF is influenced by its voltage and voltage derating, which is a very effective way to increase the PoL converter’s lifetime and reduce the capacitor’s failure rate. Nevertheless, the reasons for the capacitor’s voltage derating can have multiple causes depending on the capacitor technology and its applications. The equivalent series resistance (ESR) has become a kind of key metric for capacitors because it summarizes all the ohmic or resistive losses of the capacitor. Although tantalum capacitors provide low or good ESR performance, a new technology—conductive polymers—has been developed to provide a lower ESR value that exhibits conductivity characteristics close to those of metals. Compared to generic MnO_2_-based tantalum capacitors, this new type—using conductive polymers—is significantly more conductive (100 times), and the direct benefits of this augmented conductivity are its lower ESR, improved high-frequency behavior, and superior reliability—especially for lower-rated voltages [2]. A conductive tantalum polymer chip capacitor is a capacitor with a solid electrolyte made of a conductive polymer. In the past decade, all polymer capacitor technologies seem to have been well adopted into the PoL power electronics converter branch. They have an anode body that is made of sintered tantalum powder, followed by the dielectric—which is a thin film of TaO (generated by electrochemical oxidation)—and, finally, the cathode, consisting of a highly conductive polymer layer deposited in the oxide layer [2,3,4]. As these Tantalum Polymer devices are actually polar capacitors, it is very important when using them in practice to give attention to the polarity marking. A reverse polarity will be allowed only up to the values indicated in the data sheet. A high voltage and high current stress will eventually lead to a rapid degradation of the components or device performance. Derating designs constitute a key factor in component selection for lowering the failure rate. When performing derating, it is necessary to keep the maximum stress at a proportion of the maximum ratings—this is called the derating factor [5,6,7]. If we act in voltage, derating means that the actual capacitor shall be used in the application at a lower voltage than the rated voltage.

When they are operated within their recommended guidelines, these Tantalum polymer chip devices—which are in fact solid state capacitors—demonstrate no wear out mechanism, as we can see in bathtub curve below. Although they can be operated at the full-rated voltage, professional circuit designers look for a minimum level of assurance in long-term reliability, which should be demonstrated with data. Since most applications do require long-term reliability, a voltage derating operation can provide the desired level of demonstrated reliability, based on industry-accepted acceleration models. Many manufacturers compulsorily recommend designers to consider voltage derating for the maximum steady-state voltage. Table 1 shows the recommendations for this issue used by one important producer of capacitors.

In PoL and automotive applications—as described schematically in Figure 1—the output component position (capacitor C_out_) selection follows Table 2, depending on the mission profile and the temperature requirements. For the 12 V line, the recommendation is—at minimum—a 35 V-rated voltage capacitor, taking into consideration the existing ISO Pulse requirements defined by the ISO7637 Specification.

## 2. Materials and Methods

The operation of derating can be expressed by the percentage of the rated voltage that shall be subtracted. The aim of the derating is to reduce the number of stress factors applied to the capacitors. The two main stress factors are the voltage and temperature. The derating curve is shown in Figure 2, where V_R_ = the rated voltage, V_C_ = the rated voltage, T_lc_ = the lower category temperature, T_R_ = the rated temperature, and T_uc_ = the upper category temperature. Practically, if we carry out a 50% derating, this means that the capacitor shall be used at 50% of the rated voltage for a specific application (i.e., a 6.3 V-rated capacitor will be used on 3.15 V at maximum). Varying derating curves can be found in the MIL-HDBK-1547 military standard [8]. Due to new materials emerging on the market—corroborated by special manufacturing processes—polymer tantalum capacitors have a long way to go to meet automotive AEC-Q200 stress test requirements, and, consequently, require specific electrical transient tests too.

For the experimental part of the study, an EPC9059/30V development board containing two EPC2100 eGaN (enhancement Gallium Nitrade) FET transistors in a half bridge configuration [9,10,11,12], using the Texas Instruments LM5113 gate driver, was used in a workbench hands-on investigation (Figure 3).

The measurement connection diagram and the method for the temperature scanning are shown in Figure 4. Since the standard used for the reliability prediction required information about the capsule temperature of the component in question, an infrared thermal scanning device was used. The temperature collected by this was further used to calculate the piT stress factor. This, in turn, was used to calculate the lambda failure rate and then the MTBF. During the experiment, no forced cooling method was used for the investigated evaluation board, but only natural convection; also, the eGaN-FET transistors did not have a thermal radiator. Therefore, the extraction of information on the reliability of the converter was obtained in the most difficult conditions for thermal dissipation in active devices. The main parameters of the converter are shown in Table 3. For the DC–DC converter built with eGaN transistor technology, a tantalum polymer capacitor made with SMD technology-type encapsulation was selected for the output filter. Such converters are mission-critical in the telecom industry and computer applications, so knowledge of their capabilities in terms of their reliability is very important. This capacitor was the object of the voltage derating. In fact, two polymer capacitors—NEEgJ8, with a nominal voltage of 4 V and NEtjJ8, with a nominal voltage of 6.3 V—were used consecutively. Both were produced by NEC-Tokin Corporation headquartered in Shiroishi, Japan. (now part of KEMET). In Figure 5, these two capacitors and all the information related to their packaging, operating temperature, tolerance, identification of their technical parameters, and ESR (Equivalent Series Resistance) are shown. Additionally, on the right side of the figure, the actual values measured with a capacitor-tester for the electrical capacitance of each capacitor (255 μF vs. 220 μF, respectively) are shown, with which the calculations were made to find out the temperature stress factor *π_T_*; then, this factor was entered into the formula for devising the failure rate.

Regarding the reliability calculation method, it must be said that the standard used provides a prediction and not an estimate. Reliability prediction for electronic components is commonly based on the failure rate, which is assumed to be constant during the lifetime period in the bathtub curve, as shown in Figure 6 [8,13].
*R*(*t*) = *e*^−*λt*^(1)
where *λ* signifies the intrinsic failure rate, excluding early failures and wear-out failures. The mean time between failures (MTBF), i.e., when the reliability function has decayed to a value of *e*^−1^ = 36.7% (or in other words, only 37% of the units within a large group will last as long as the *MTBF* number) is: *MTBF* = *λ*^−1^(2)

The International Electrotechnical Commission released the IEC-TR-62380 standard in 2004, including features of newer components arriving on the market such as polymer capacitors, which did not appear in older standards such as MIL-HDBK-217 [8]. Thus, we performed the calculation of the capacitor’s *MTBF* based upon the IEC-TR-62380 standard, in which a mathematical model for every electronic component is defined, in order to calculate their failure rates *λ* [13]. The reliability data taken from this prediction standard are taken mainly from field data concerning electronic equipment operating in that type of environment*: Ground; stationary; weather-protected* (means: equipment for stationary use on the ground in weather-protected locations, operating permanently or otherwise with controlled temperature and humidity and good maintenance). This applies mainly to telecommunications equipment and computer hardware. Experience has shown that component reliability is heavily influenced by mechanical and environmental conditions, as well as by electrical environment conditions. Estimated reliability calculations of equipment have to be carried out according to its field use conditions, so they are defined by the mission profile. A “mission profile” is usually defined as a table that includes all the details on the ambient temperature cycles during the lifetime of the device being discussed, on/off—state durations, numbers of operation cycles, etc. [8,13].

## 3. Results

### 3.1. Experimental Workbench SET-UP, Key Measurements, and the Mode of Investigation

An experimental workbench hands-on SET-UP was constructed in order to obtain a synchronous buck converter that lowers its voltage from 12 V to 1.2 V (a very often-used amount of voltage for powering FPGAs, microprocessors, etc.). The actual Capacitance, ESR, and V_loss_ (%) values for the two capacitors were measured using an BSIDE^®^ *ESR02 Pro* transistor tester. The output filter reliability was improved by putting a higher-voltage rating capacitor on the output rail in parallel to the load resistance. Then, a thermal scan of the surface of the capacitor was executed in order to determine the temperature of operation, which is necessary in the calculation of the failure rate *λ*. In Figure 7 and Figure 8 the thermal picture of the evaluation board area is shown, scanned by mean of a thermo-vision camera—specifically, the temperature of the capacitor’s capsule was 36 °C for the 4 V-rated capacitor and 36 °C for the 6.3 V capacitator. These situations indicate to us that in the case of a derated capacitor, a smaller amount of current passes through it (i.e., a decreased power loss occurs). Later, using this information, the failure rate was calculated using the temperatures obtained and the models provided by the prediction standard chosen for the investigated capacitors. Finally, the MTBF was determined—a strong indicator of overall reliability. The scan process also showed that the highest temperature on the EVB is attributed to the coil, so this component contributed the most to decreases in the general reliability of the studied converter.

This is in agreement with the Arrhenius law (Figure 9), which stipulates that if the capacitor’s temperature increases by 10 °C, then the reliability will decrease by half. 

The good operation of the converter was experimentally validated by measuring some important waveforms, such as:The intensity of the current through the converter’s inductance L = 3 μH (a SMD power coil constructed with flat wire windings), which was collected by a current probe. The current probe used in the experiment had a 10-mV output voltage, corresponding to 4 A as measured.The voltage measured in the so-called “switching point” (at the junction between high FET and synchro FET).

The most important waveforms measured on the converter under load at room temperature are shown in Figure 10. These waveforms ensure the CCM mode’s functioning in the converter—i.e., Continuous Conduction Mode. A general view of the experimental set-up is shown in Figure 11. 

### 3.2. Reliability Calculation

In order to execute the calculation for the tantalum polymer capacitor, the mathematical model for this type of capacitor was taken from the IEC-TR-26380 standard [13], as shown in Figure 12.
*λ* = 0.4 × {[(*Σ*(*π_T_*)*_i_* × *τ_i_*)/(*τ_on_* + *τ_off_*)] + 3.8 × 10^−3^ × [*Σ*(*π_n_*)*i* × (Δ*T_i_*)^0.68^]} × 10^−9^/h(3)

The Standard specifies that the above formula gives field values if the ratio peak voltage/rated voltage is less than or equal to 0.8 (with peak voltage = continuous voltage + peak value of the alternative voltage). In the investigated circuit, the peak voltage was 1.5 V and the maximum rated voltage for the capacitors was 6.3 V, so the ratio was equal to 0.238 < 0.8. 

After performing some preliminary calculations, which we will not reproduce here as they are irrelevant, it was determined that:*τ_i_* = 365 days × duty cycle = 365 × 0.13*π_T_* = 0.9 derived from Figure 11 @ 25 °C*n_i_* = 365 and (*π_n_*)*_i_* = (*n_i_*^0.76^) × 1.7*τ_on_* = 1 and *τ_off_* = 0*Σ*(*π_n_*)*_i_ = n_i_* and *Σ*(*π_T_*)*_i_ = π_T_*Δ*T_i_* = capacitor’s surface temperature—ambient temperature
where Δ*T_i_* is derived using the actual temperature of the capacitor’s capsule and *π_T_* from Figure 13.

Figure 13 shows the temperature factor at an ambient temperature of ~25 °C, at which the experimental measurements took place. Many details are given in subchapter 8, mission profiles, within the IEC-TR-62380 standard [13]—hence: *λ_polimer capA_* = 0.4 × {[(0.9 × 54.75)/(1 + 0)] + 1.4 × (10^−3^) × [(1.7 × 365^0.6^) × (36 − 25)^0.68^]} × (10^−9^)/h = 7215.425 FIT = 7.215425 F/10^6^ h*λ_polimer capB_*[derated] = 0.4 × {[(0.9 × 54.75)/(1 + 0)] + 1.4 × (10^−3^) × [(1.7 × 365^0.6^) × (31 − 25)^0.68^]} × (10^−9^)/h = 6821.755 FIT = 6.821755 F/10^6^ h
where FIT stands for Failures in time (1 FIT = one failure in 10^9^ h) and F/10^6^ h stands for Failures in one million hours.

Consequently, we have:*MTBF_polimer capA_* = 138,591 h or 15.821 yr*MTBF_polimer capB_*[derated] = 146,589 h or 16.734 yr

The failure rate in FIT (failures per 10^9^ h) and the MTBF in hours for the two investigated capacitors are presented in a suggestive comparative diagram, shown in Figure 14.

## 4. Discussion

Derating the stress levels of electronic components is recognized as a very effective way of improving the device’s reliability and also for avoiding failures due to overstress. This paper provides a practical study that refers to a simple way of augmenting the reliability of the output filter of modern eGan transistor-based DC–DC converters. Experimental results combined with calculations based on data and models provide by the IEC-TR-62380 reliability prediction standard were validated by performing a voltage derating operation upon the converter’s output capacitor, resulting in an improvement in the MTBF of about 7%. This was possible using a 6.3 V-rated polymer tantalum capacitor instead of a 4 V-rated capacitator on a 1.2 V-output voltage rail. 

## Figures and Tables

**Figure 1 micromachines-14-00221-f001:**
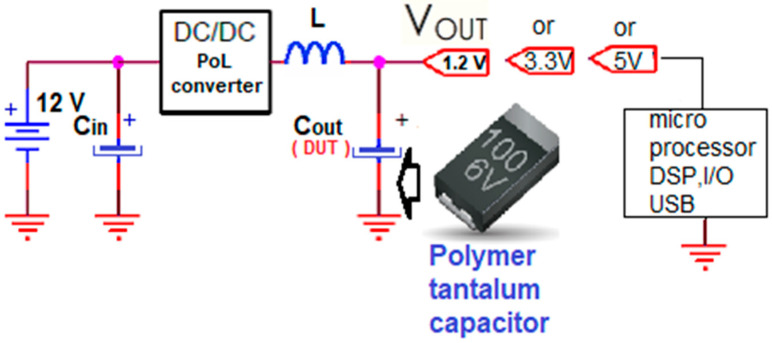
Schematic diagram for PoL and Automotive Applications.

**Figure 2 micromachines-14-00221-f002:**
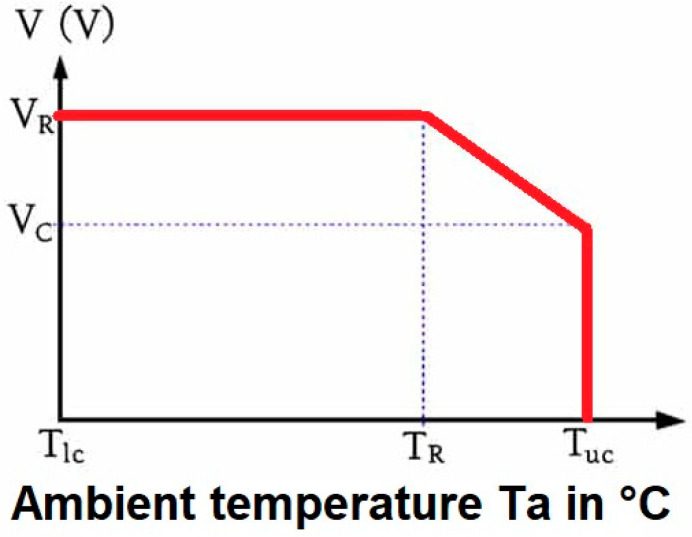
The derating curve.

**Figure 3 micromachines-14-00221-f003:**
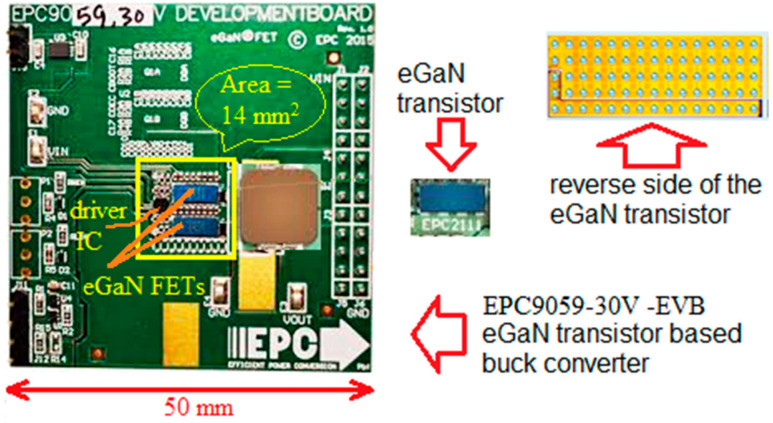
Planted area view of the eGaN evaluation board EPC9059-30V used in the experiment, with dimensions; the eGan-transistor’s area is outlined in yellow and the reverse side of the eGaN transistors (EPC2111) has a Ball Grid Array (BGA packaging technology) [10].

**Figure 4 micromachines-14-00221-f004:**
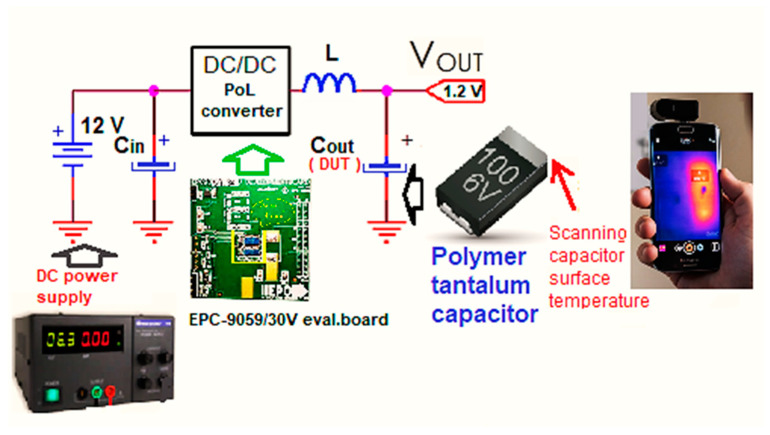
Diagram of connections with the derated polymer tantalum capacitors and how the capacitor’s temperature is detected.

**Figure 5 micromachines-14-00221-f005:**
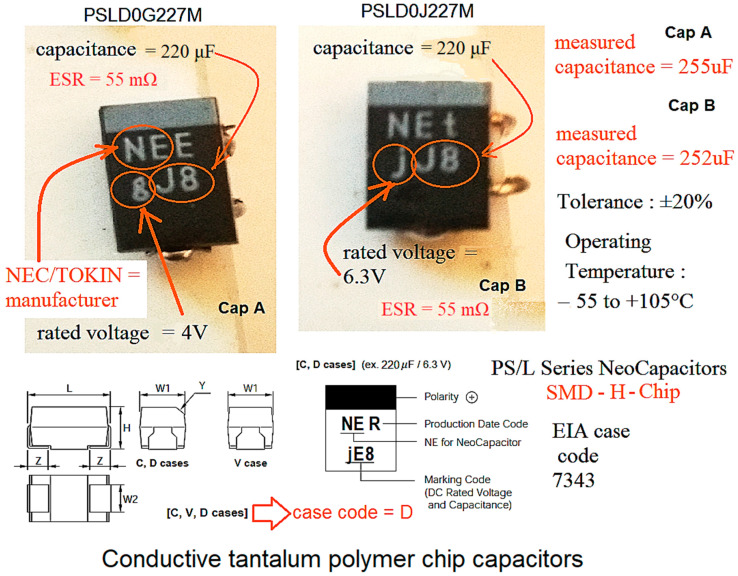
Packaging and identification information for the SMD polymer capacitors used for the reliability investigation of the eGan converter.

**Figure 6 micromachines-14-00221-f006:**
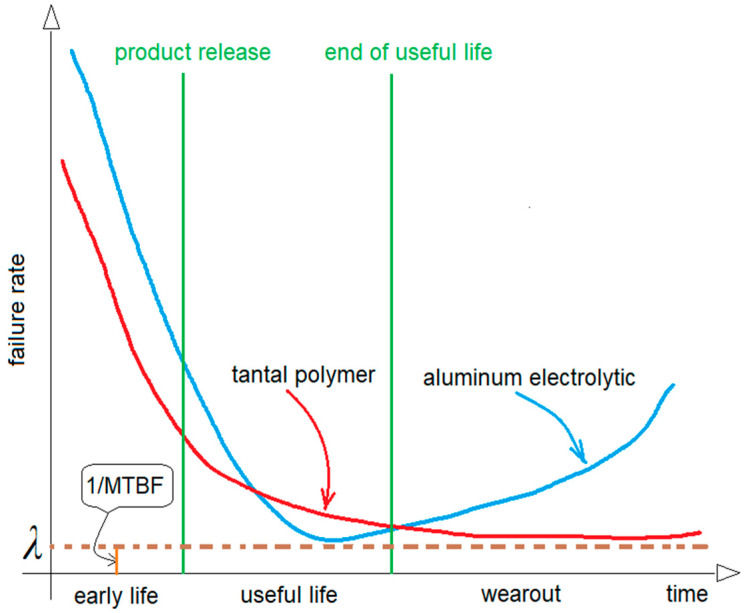
The bathtub curve.

**Figure 7 micromachines-14-00221-f007:**
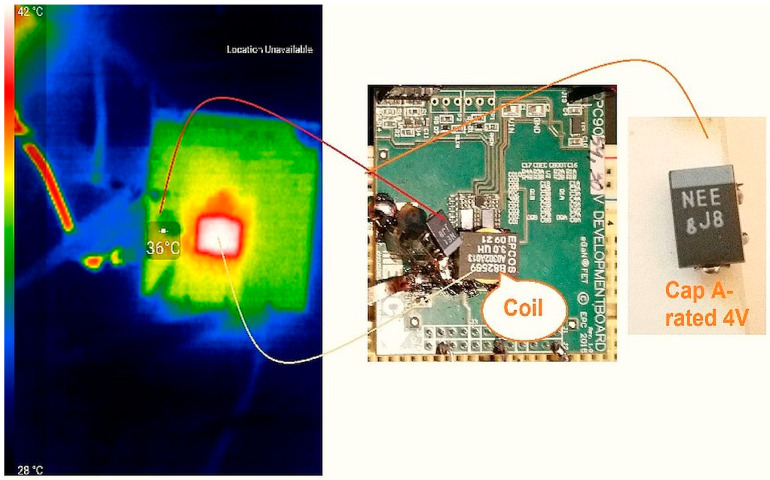
Thermal image of the Capacitor A surface. A 36 °C temperature is in evidence, as shown by the infrared thermo-vision device. Left—EVB scanned in infrared. Middle—EVB seen from the side, plated with components. Right—Photo of the investigated capacitor.

**Figure 8 micromachines-14-00221-f008:**
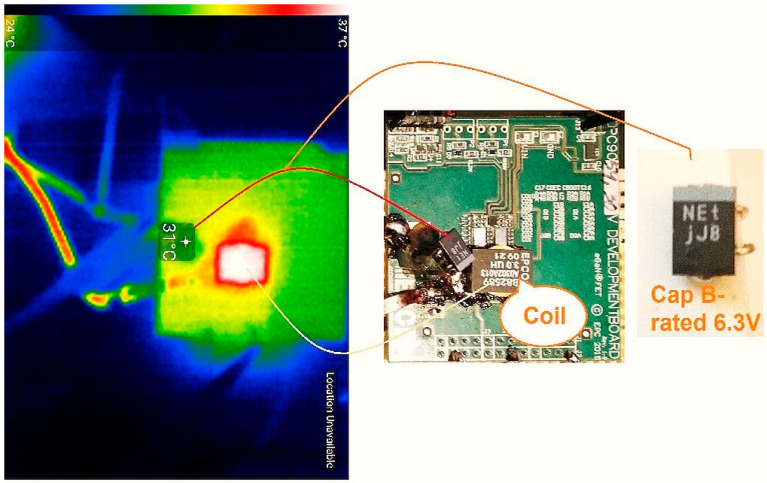
Thermal image of the Capacitor B surface (derated). A 31 °C temperature is in evidence, as shown by the infrared thermo-vision device. Left—EVB scanned in infrared. Middle—EVB seen from the side, plated with components. Right—Photo of the investigated capacitor.

**Figure 9 micromachines-14-00221-f009:**
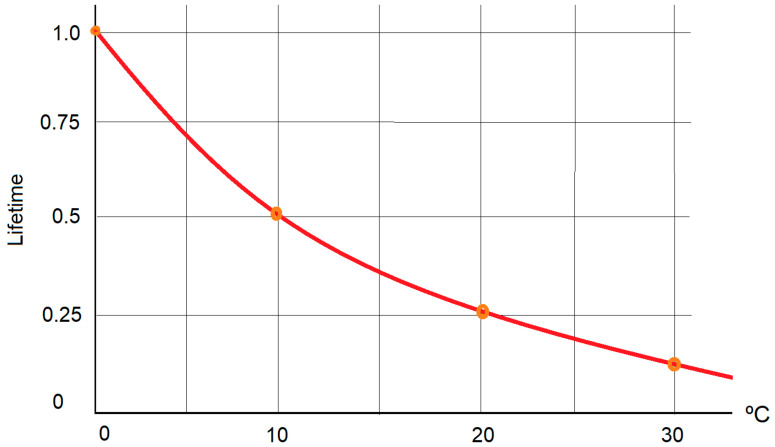
Arrhenius Law.

**Figure 10 micromachines-14-00221-f010:**
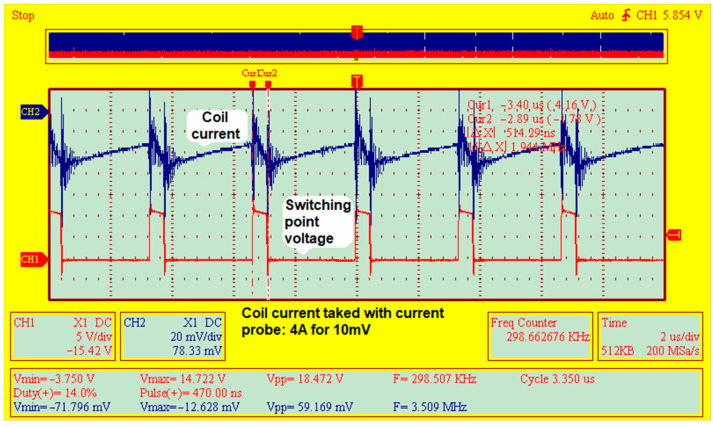
Basic waveforms of the PoL DC–DC eGaN-FET-based converter, ensuring the CCM mode of operation.

**Figure 11 micromachines-14-00221-f011:**
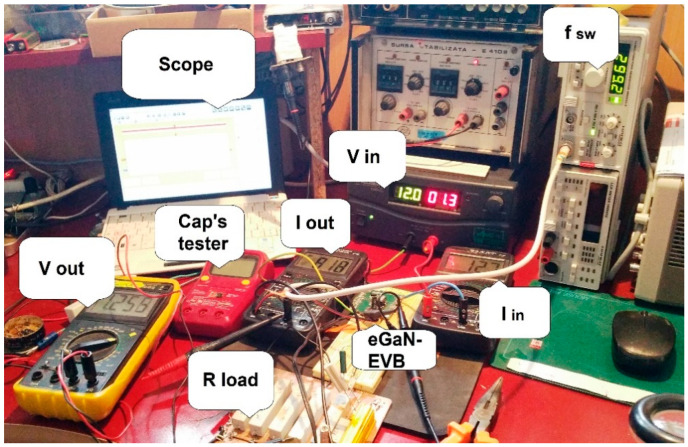
General view of the experimental set-up. Voltage waveforms were displayed using an USB-DSO oscilloscope—ISDS 220B and a CP-05^+^ current probe.

**Figure 12 micromachines-14-00221-f012:**
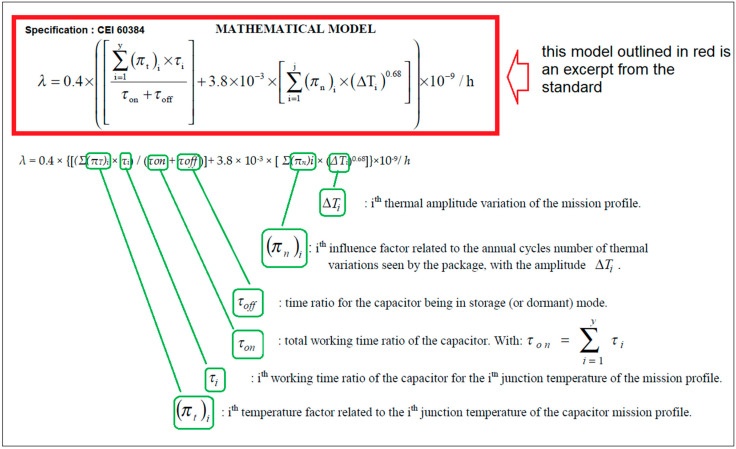
Explanation of how the failure rate λ is devised using the models for Tantalum Polymer Capacitors (the models are provided by the IEC-TR-62380 Standard). The mathematical model and parameter’s formulae, outlined in green, are depicted as they appear in this standard.

**Figure 13 micromachines-14-00221-f013:**
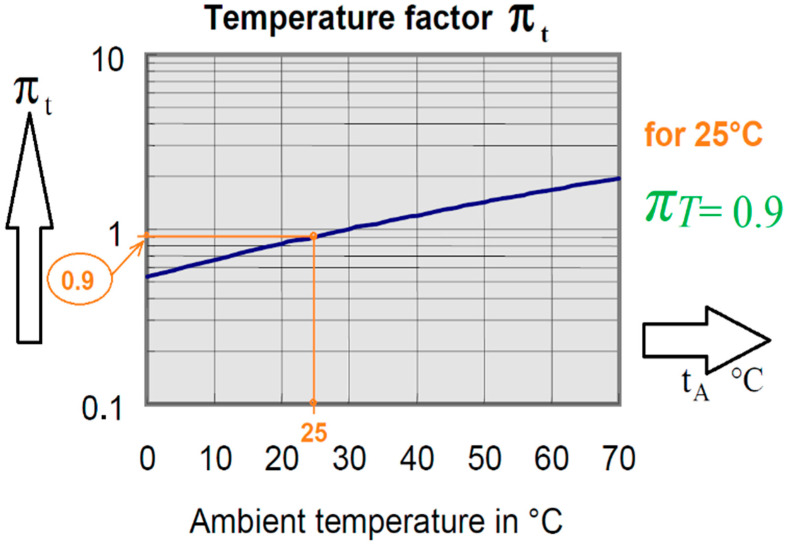
Temperature factor *π_T_* versus ambient temperature for the tantalum polymer capacitor (graphical derived—orange lines—using a nomogram from the IEC-TR-62380 standard at 25 °C [13]).

**Figure 14 micromachines-14-00221-f014:**
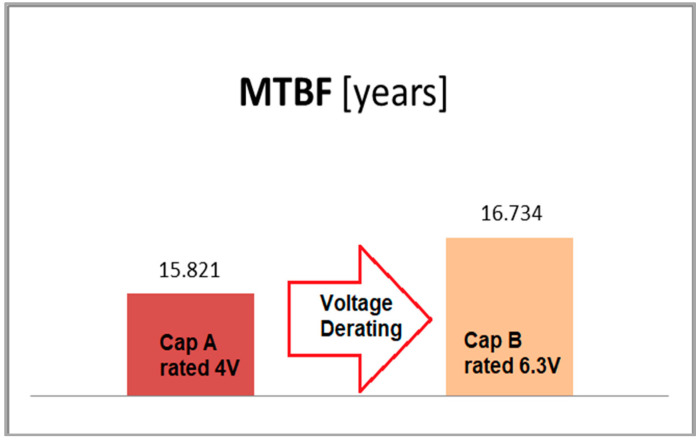
Graphical comparison of the MTBF for the two capacitors.

**Table 1 micromachines-14-00221-t001:** Capacitor’s derating for the range 2.5 V to 35 V Rated voltage; Above 105 °C, the voltage is de-rated linearly to 0.67 × Rated voltage up to the maximum operational temperature.

Rated Voltage		Derated Voltage	
−55 °C to 05 °C	−55 °C to 125 °C	−55 °C to 105 °C	−55 °C to 125 °C
2.5 V	1.7 V	2.3 V	1.5 V
4 V	2.7 V	3.6 V	2.4 V
6.3 V	4.2 V	5.7 V	3.8 V
10 V	6.7 V	9 V	6 V
16 V	10.7 V	12.8 V	8.6 V
20 V	13.4 V	16 V	10.7 V
25 V	16.8 V	20 V	13.4 V
35 V	23.5 V	28 V	18.8 V

**Table 2 micromachines-14-00221-t002:** Capacitor’s Application Voltage—Recommended Capacitors.

Application Voltage	Mission Profile Temperature (up to 105 °C)	Mission Profile Temperature (up to 125 °C)
<1 V	2.5 V	2.5 V
3.3 V	4 V	6.3 V
5 V	6.3 V	10 V
12 V	Minimum 35 V	Minimum 35 V

**Table 3 micromachines-14-00221-t003:** Parameters for the investigated converter.

Parameter	Value
Load resistor	*R_load_* = 0.1 Ω
Load current	*I_out_* = 12 A
Input current	*I_in_* = 1.2 A
Input voltage	*V_in_* = 12 V
Output voltage	*V_out_* =1.2V
Switching frequency	*f_sw_* = 300 kHz
Inductor	*L* = 3 μH
Ambient temperature	~25 °C
Duty cycle	~13%
Output *C_tantalum-poly_*_A_	220 μF/4 V
Output *C_tantalum-poly_*_B_ [derated]	220 μF/6.3 V

## Data Availability

Not applicable.

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
