# Peer review of "A Derating-Sensitive Tantalum Polymer Capacitor’s Failure Rate within a DC-DC eGaN-FET-Based PoL Converter Workbench Study"

_micromachines, 2023, doi:10.3390/mi14010221_

Round 1

Reviewer 1 Report (Previous Reviewer 1)

I checked the letter and the manuscript and believe that the manuscript has been significantly improved and now can be published in Micromachines.

Reviewer 2 Report (Previous Reviewer 2)

good revisions 

This manuscript is a resubmission of an earlier submission. The following is a list of the peer review reports and author responses from that submission.

Round 1

Reviewer 1 Report

Corrections needed:

Title: "... Polymer Capacitor's ..." 

Intro: "A polymer capacitor is an electrolytic capacitor with a solid electrolyte made of conductive polymer".

Text: ... "a polymer capacitor made in multilayer technology"

Polymer Tantalum capacitors (not Polymer Capacitors) aren't electrolytic; conductive polymer cathode in these capacitors isn't solid electrolyte; technology of these capacitors isn't multilayer technology.

The info on science, technology and applications of Polymer Tantalum capacitors can be found in Y. Freeman, Tantalum and Niobium-Based Capacitors, Springer, 1st edition 2017, 2nd edition 2022

Reviewer 2 Report

1. Concept of the paper is good and well explained.

2. Abstract to be modified in a better way about the paper.

3. Please avoid self-citation which are not required. 

4. Figure.2 & 3 need to be modified and should be more clear.

5. In page no 4 there are two figures mentioned with same Figure no as 4 please clarify and clarity in DPI is required.

6. In page 6 figure 6 is copied from the internet source please redraw.

7. Figure 9 & 10 is looks like a text book figure there is no naturality in the paper.

8. Authors can be explained in a better way about experimental analysis and there is lot of misconception in the paper which need to improve.